# The effect of location in mining or borderland areas on HIV incidence among people who use drugs attending a harm reduction programme in Myanmar, 2014– 2021: A retrospective cohort study

**Lucy Platt**[1]*, **Khine Wut Yee Kyaw**[2], **Sujit D. Rathod**[1], **Aung Yu Naing**[2], **Sophia Garkov**[1], **Murdo Bijl**[2], **Bayard Roberts**[1]

**1** London School of Hygiene and Tropical Medicine, London, United Kingdom, **2** Asian Harm Reduction Network, Yangon, Myanmar

\* lucy.platt@lshtm.ac.uk

## Abstract

### Background

High HIV prevalence has been documented among people who inject drugs in Myanmar particularly in mining and borderland areas. We estimated incidence of HIV among people using drugs (via injecting and other routes) and examine associations between location in mining or borderland areas and risk of infection.

### Methods and findings

Analysis of data among clients registered at harm reduction programmes across Sagaing region, Kachin, and Northern Shan States between 2014–2021. Data on socio-demographic, drug use characteristics and clinic-level data on borderland or mining locations were collected at time of registration. Characteristics, repeat HIV testing and HIV seroconversion were analysed using a cohort approach. We use Poisson regression models to examine associations between location in a borderland or mining area and incidence of HIV, adjusting for confounders. Data were available for 85,093 clients, 52,526 reported HIV tests and 20.0% were seropositive. 38,670 clients had no or only one recorded HIV result. The median time between HIV tests was 1.1 years. Among 13,359 clients with 2 or more HIV tests the HIV seroconversion rate was 3.8 per 100 person years (pyrs) (95% CI 3.6–4.0). Incidence among those who injected drugs was 6.8 per 100/pyrs, 8.9 among those aged $\leq$ 25 years, 2.3 among women, 2.3 among those who had migrated, 5.6 among those located in border areas, and 3.7 among those in mining areas. After adjusting for confounders, HIV incidence remained higher for people located in borderland areas (Incidence Rate Ratio 1.67 95% CI 1.13–2.45) but there was no evidence of association between location in a mining area and HIV seroconversion.

**Data availability statement:** Data cannot be shared publicly because the dataset contains individual level information on a population engaging in illegal activities (drug use) in a country currently led by a military dictatorship. The dataset contains data on demographic characteristics and geographical location of individuals that taken together could lead to deductive disclosure of individuals. If required a restricted dataset could be make available on request to AHRN the data holders following the implementation of a data sharing agreement and ethics approvals. Please direct requests to contact@ahrnmyanmar.org Postal address: No. 135(G), Mawyawaddy Street, Pyay Road, 8 Mile,Yangon

**Funding:** This work is supported by the Wellcome Trust [226619/Z/22/Z] and the ESRC, Drugs and (dis)order: Building sustainable peacetime economies in the aftermath of war, UKRI Award no. ES/P011543/1, 2017–2021, as part of the Global Challenges Research Fund. The funders had no role in the design of the study, collection, analysis, interpretation of data and writing of the manuscript. There was no additional external funding received for this study.

## Conclusions

Findings highlight the need to intensify harm reduction interventions with a focus on cross-border interventions. Increasing uptake of HIV testing alongside the scale up of evidenced based interventions is urgently needed to curb the high rates of HIV transmission associated with drug use, particularly among young people.

## Introduction

An estimated 4 million people who inject drugs live in East and South-East Asia, representing between 20% and 30% of the global population [1]. Use of amphetamine type stimulants (ATS) is increasing, an estimated 20 million people are using ATS globally, and use and production is prominent in South East Asia [2]. Myanmar is the largest producer of ATS in the South East Asian region and the world's second largest producer of opium [3]. Increased availability of drugs in producer countries or along trafficking routes combined with political instability and reduced enforcement has been linked to elevated drug use and outbreaks of HIV infection [4,5]. Within Myanmar, drug production areas such as Kachin State, Northern Shan State, and the Sagaing region have the highest prevalence of drug use in the country [6].

While HIV incidence among people who inject drugs is declining in Europe and North America, in many parts of South East Asia, Russia and Eastern Europe it is increasing, contributing between 30 and 39% of all new HIV infections in 2021 [1,7,8]. A recent retrospective cohort study estimated incidence among the population in Kachin State, Myanmar to be 7.1 per 100/pyrs, declining from 19.1 in 2008–2011 to 5.2 in 2017–2020 [9]. This is far higher than the global incidence of 2.53 per 100 person years [8]. There is a growing body of evidence documenting high HIV prevalence (34.9%) among people who inject drugs in Myanmar ranging from 7.6% to 61% and higher in rural areas of Bhamo and Waingmaw (61–56%) in Kachin State [10]. Less is known about the prevalence of HIV among people using drugs through non-injecting routes both globally and in Myanmar. HIV risk depends on levels of unprotected sex within the population, engagement in sex work, the extent of sexual networks across injecting and other drug using communities as well as rates of transition between injecting and non-injecting practices [11]. In Vietnam, HIV prevalence was 6.3% among young people (15–24 years) predominantly using methamphetamines and higher (15%) among those with a history of injecting [12]. In Myanmar, people using methamphetamines in Shan State frequently reported inconsistent condom use and multiple sex partners pointing to the potential for sexual transmission of HIV [5].

The 'risk environment' concept, developed to understand drug-related harms examines different types (physical, social, economic and political) and levels of influence (e.g., individual, community or national) in line with broader efforts to address structural determinants of health [13,14]. Neighbourhoods or residential areas at a community level can have specific political, physical and social attributes that affects health of individuals [15]. Research in urban areas indicates that neighbourhood income is linked to higher rates of fatal overdoses among people using opioids with more overdoses occurring in fragmented lower income neighbourhoods. Physical aspects of the built environment such as run-down buildings are linked to higher incidence of drug use and overdose mortality [16]. Less is known about rural risk environments, but limited harm reduction services in rural areas has been linked to higher risk of overdose in the United States [17].

Borderland areas in Myanmar are a central hub in the global illegal drug trade, with research suggesting higher HIV prevalence, increased drug use and harms including more frequent injecting, equipment sharing practices and reduced access to harm reduction

services [10,18,19]. Borderlands and other rural areas in Kachin and Shan states and Sagaing region are one of the world's largest sources of jade, tin and rare earth mines, as well as gold, amber and coal [20]. Qualitative research points to the widespread availability of drugs in mines, with workers often paid in opium and employers supporting drug use, to promote harder working and tolerance of dangerous conditions [5,21,22]. Both areas have high levels of population mobility, with people working in agriculture or sex work around mining communities or as a result of forced displacement or cross-border movement for work [5]. The opening of mines creates economic opportunities in the area and the establishment of communities. Research suggests lower levels of risk aversion in relation to drug use and sexual practices within these communities linked to working away from home or work in dangerous conditions [23]. Mountainous terrains in borderlands are conducive for drug production and facilitate undetected distribution across borders, but also present physical barriers to education and health services [24]. These physical and social attributes highlight borderland and mining areas as important aspects of the risk environment.

To date, epidemiological research among people who use drugs in Myanmar has focused on understanding individual risk practices, and there is limited evidence on the extent to which economic, legal, social and political factors shape HIV acquisition despite widespread recognition of their importance in understanding and reducing risk of infection [25,26]. We undertook an analysis of programme data from a harm reduction service in Myanmar with the aim of estimating the incidence of HIV among people who use drugs (both through injecting and other routes) and examine the extent to which location in borderland and mining areas affect risk of infection.

## Methods

### Setting

Myanmar has suffered protracted armed conflict leading to large-scale forced displacement with millions moving within Myanmar as internally displaced persons or into neighboring countries as refugees. A total of 912,000 people are internally displaced across Myanmar, with the largest populations in Kachin, Chin, Shan and Rakhine [27]. Sagaing region, Kachin and Northern Shan state have vast borderland areas adjoining India, China, Laos and Thailand, primarily composed of marginalized ethnic groups, including ethnic armed groups and military activity. Mining, illegal drug production and ethnic conflict are obstacles to provision of effective HIV response in these borderland states [18]. The Asian Harm Reduction Network (AHRN) has been providing harm reduction services in Northern Shan state since 2003 and expanded to Kachin State and Sagaing region under the National AIDS Programme guidelines for the treatment and prevention of HIV among key populations. Attendance is voluntary and services include provision of needles and syringes; condom and lubricant distribution; opioid agonist therapy (OAT); HIV testing and counselling; antiretroviral therapy (ART) provision, information, education and communication; Hepatitis C and B testing and treatment; Hepatitis B vaccination; sexually transmitted infection (STI) and Tuberculosis (TB) prevention, diagnosis and treatment; and mental health assessment, treatment and referral. Aluminium foil is distributed for smoking of opium or heroin to encourage transition away from injecting or to reduce sharing smoking equipment. Clients consist of people who use or inject drugs, their sexual partners and family members. For this analysis we focus on people who currently use drugs. We focus on individuals: i) with one or more HIV test results; ii) who tested HIV negative at first HIV test; and iii) who registered since 2014 when ART scale-up was intensive and routine HIV testing was increased.

## Study design and data collection

We conducted longitudinal analysis of routine data collected from 35 of AHRN's project sites across 22 townships in Myanmar between January 2014 and December 2021. Project sites include fixed sites (drop-in centres) and through mobile teams to reach the largely rural populations. New clients are provided with a unique identifier and complete a standardized registration form. This is completed on paper at any service (mobiles, via outreach, at drop-in centres (DIC)) and then entered into an electronic database. Questions on drug use (type and mode) and other demographic characteristics are recorded during registration.

HIV testing is provided at fixed site DICs and through mobile medical teams with a suggested frequency of six months. HIV testing is encouraged through outreach workers and via peer workers. Counsellors (nurses or peers) provide pre-test counselling, and after obtaining consent, whole blood specimens (finger prick or venipuncture) are taken in DICs or mobile clinics and tested using Alere Determine HIV 1–2 (Alere Medical Company Ltd, Japan). Reactive samples are retested at laboratories (for DICs) or at DICs (following community testing) using confirmatory tests conducted in parallel Unigold (Trinity Biotech Manufacturing Ltd., Ireland) and Stat-pak (Chembio Diagnostic Systems Ltd., USA) with post-test counselling provided at return of results. Clients with confirmed HIV positive results are referred to AHRN ART satellite sites for pre-ART assessment, co-trimoxazole preventive therapy, opportunistic infections screening, treatment and counselling.

## Covariables

Key exposures were indicators of geographical context including: (i) location of clinics in borderland; or (ii) mining areas, conceptualised as significant aspects of the risk environment with specific physical and social attributes linked to elevated risk of HIV transmission. Location in borderland or mining areas were extracted from profiles of townships, crossed checked with project staff and attributed to individuals according to their location at registration (at DIC or outreach) [28]. Borderland areas are situated along the borders with China and India with the presence of cross-border trade and movement. Mining areas are identified based on the presence of significant mining activities, particularly jade, gold mining. AHRN staff conduct comprehensive mapping exercises to delineate the specific boundaries of borderland and mining areas. Clinics were classified as being in the states of Kachin, or Shan (North) or Sagaing Region.

We considered other factors associated with HIV risk including gender (male/female); education (illiterate, primary/literate, completed secondary and tertiary or more); age (≤25, 25–34, 35–44, ≥45 years); marital status (single, married, widowed/divorced); history of injecting (yes/no). Experience of migration was self-reported defined as living away from a hometown and movement to various locations for three months or more. Drug use (current) was grouped into heroin (yes/no), ATS, other drugs (including alcohol and methadone maintenance therapy), with clients being able to report multiple drugs. Occupation was recoded from an open-text response into nine categories. We defined a missing data category for covariables with >10% missing data. All indicators represent characteristics reported at the time of client registration.

## Outcome

HIV seroconversion was defined as a positive test result on a date after a negative result, per AHRN clinical records. We included clients who had an HIV negative test result followed by one or more completed test. Applying methods for estimating HIV incidence using routine clinic records [29], we calculated HIV-negative survival time for each client, starting with the

registration date, when exposures and covariable data were collected. Survival time for clients who remained seronegative ended on the date of their last HIV negative test result and for those that seroconverted at the midpoint of their last HIV negative result date and their first HIV positive result date.

## Data analysis

We describe the client population of people who use drugs served by AHRN between January 2014 and September 2021 stratified by exclusion criteria (no HIV test; testing HIV positive at first test; and only 1 HIV test) in order to assess potential selection bias of the sample included in the HIV incidence analysis. We use medians and IQRs for continuous variables and counts and percentages for categorical variables.

We present the sample included in the HIV incidence analysis stratified by our two exposures of interest (location in borderland or mining area). We estimated crude incidence rate ratios for our exposures and other covariables on HIV incidence using separate univariable Poisson regression models. Finally, we estimated incidence rate ratios for the exposures, using separate multivariable Poisson regression models with adjustment for potential confounders. All models adjusted for age, education, and registration year. For the effect of being in a border area, we adjusted for gender, state of clinic, and mining catchment area and for being in a mining catchment area, we adjusted for gender additionally. Confounders were selected based on the available covariables in the dataset which were imbalanced by exposure status, were potential risk factors for HIV, and unlikely to be on the causal pathway between the exposure and outcome. We did not adjust for marital status, migration, or occupation due to the high level of missing data for these measures. Poisson regression models included the log survival time as an offset, and had standard errors adjusted for project sites (mobile and DIC) as clusters. Participants with missing exposure data were excluded from analyses involving that exposure. We used Stata 18 (Stata Corp) in all analyses.

## Ethics

Ethical approval was obtained from LSHTM research ethics committee (Ref: 22838).

## Results

Between January 2014 and September 2021, there were 85,093 people registered across 35 AHRN project sites and 22 townships (Table 1). The median age was 38 years, 95.6% were male, 4.6% had not completed any school or were illiterate, and 18.4% were married. While 15.8% identified as having experience of migration, another 41.5% had migration status missing. Occupation was diverse, 32.0% worked in agriculture, 24.3% in mining industries and 6.0% were students or had no income, 25.3% of occupation status was missing. Heroin was used by 93.9% of clients, 59.8% used ATS, and 48.9% used opium. Overall, 48.5% injected drugs. For service registration location, 19.5% were registered in a border area while 76.2% were registered in a mining area. We excluded 71,734 clients from the HIV incidence analysis for the following reasons: no HIV test (n=32,567, 38.3%), lack of a follow-up HIV test (n=27,040, 32.8%), initial HIV positive result (n=11,630, 13.7%), and due to inability to match names to HIV results or discrepancies between test date and registration data (n=497, 0.5%) (Fig 1).

The median age of included participants was younger than the total sample (36 vs 38 years) and those people who had no HIV test were marginally younger (36 years). Proportionally more participants reporting migration had only one HIV test compared to the total sample included (20.% vs 15.8%). There were higher levels of opium use among included participants compared to the total sample (55.1% vs 48.9%) and ATS use (64.7% vs 59.8%). Proportionally

**Table 1. Characteristics of newly registered AHRN clients by inclusion status, Myanmar, 2014-2022.**

| Variable | Total | Excluded participants* | | | Included |
|---|---|---|---|---|---|
| | | No HIV test | HIV+ | One HIV test | |
| Total n(%) row | 85093 (100) | 32567 (38.3) | 11630 (13.7) | 27040 (32.8) | 13359 (15.7%) |
| | n(%) col | n(%)col | n(%)col | n(%)col | n(%)col |
| HIV positive | 11630 (16.2) | 32567 (100) | 11630 (100) | 0(0) | 0(0) |
| Age, median (IQR) | 38 (31-46) | 36 (30-43) | 33 (28-40) | 36 (29-44) | 36 (30-43) |
| <=25 | 6683(7.8) | 2091 (6.4) | 944 (8.1) | 2709 (10.0) | 693 (5.2) |
| 25-34 | 32550 (38.2) | 12565 (38.6) | 5647 (48.6) | 9887 (36.6) | 4287 (32.0) |
| 35-44 | 27256(32.0) | 10782 (33.1) | 3594 (30.9) | 8077 (29.9) | 4648 (34.8) |
| >=45 | 18648 (21.9) | 6989 (21.5) | 1444 (12.4) | 6340 (23.4) | 3730 (27.9) |
| **Gender** | | | | | |
| Male | 81314 (95.6) | 31229 (95.9) | 11323 (97.4) | 25553 (94.5) | 12736 (95.3) |
| Female | 3779 (4.4) | 1338 (4.1) | 307 (2.6) | 1487 (5.5) | 623 (4.6) |
| **Education** | | | | | |
| Illiterate/other | 3909(4.6) | 1386 (4.3) | 501 (4.3) | 1517 (5.6) | 498 (3.7) |
| Primary/literate | 24258 (28.5) | 8051 (24.7) | 3406 (28.3) | 8102 (30.0) | 4511 (33.8) |
| Up to secondary | 43660 (51.3) | 15124 (46.4) | 6764 (58.2) | 14345 (53.0) | 7173 (53.7) |
| Tertiary or more | 3283 (3.8) | 1253 (3.8) | 311 (2.7) | 1141 (4.2) | 556 (4.2) |
| Missing | 9983 (11.7) | 6753 (20.7) | 648 (5.6) | 1935 (7.2) | 621 (4.6) |
| **Marital status** | | | | | |
| Post-marital | 3024 (3.5) | 576 (1.8) | 614 (5.3) | 1249 (4.6) | 555 (4.1) |
| Married | 15684(18.4) | 3078 (7.4) | 1738 (14.9) | 7431 (27.5) | 3293 (24.6) |
| Single | 11271 (13.2) | 2221 (6.8) | 2118 (18.2) | 5060 (18.7) | 1780 (13.3) |
| Missing | 55114(64.7) | 26692 (82.0) | 7160 (61.6) | 13300 (49.2) | 7731 57.8) |
| **Occupation** | | | | | |
| Agriculture | 27240 (32.0) | 8819 (27.1) | 4078 (35.1) | 9486 (35.1) | 4631(34.7) |
| Driver | 3291 (3.9) | 1421 (4.4) | 324 (2.8) | 985 (3.6) | 536 (4.0) |
| Drug/casino/sex | 1239 (1.5) | 406 (1.2) | 103 (0.9) | 446 (1.6) | 270 (2.0) |
| Mining | 20642 (24.3) | 6616 (20.3) | 3267 (28.1) | 6874 (25.4) | 3798 (28.4) |
| Student/ no income | 5110 (6.0) | 1923 (5.9) | 828 (7.1) | 1604 (5.9) | 735 (5.5) |
| Casual work | 3758 (4.4) | 1547 (4.7) | 499 (4.3) | 1081 (4.0) | 602 (4.5) |
| Uniformed officer | 17 (0.0) | 6 (0.2) | 3 (0.03) | 8 (0.03) | 0 (0.0) |
| Skilled/ office | 2240(2.6) | 830 (2.5) | 279 (2.4) | 795 (2.9) | 326 (2.4) |
| Other/ missing | 21556 (25.3) | 10999 (33.8) | 2249 (19.3) | 5761 (21.3) | 2461 (18.4) |
| **Experience of migration** | | | | | |
| No | 36311(42.7) | 13793 (42.3) | 4569 (38.3) | 12257 (45.7) | 5319 (39.8) |
| Yes | 13426 (15.8) | 4229 (13.0) | 1559 (13.4) | 5541 (20.5) | 2043 (15.3) |
| Missing | 35356 (41.5) | 14546 (44.7) | 5502 (47.3) | 9142 (33.8) | 5997 (44.9) |
| **Drug use (yes vs no)** | | | | | |
| Heroin | 79930 (93.9) | 30547 (93.8) | 11437 (98.3) | 24569 (90.9) | 12897 (96.5) |
| Opium | 41600 (48.9) | 14375 (44.1) | 6180 (53.1) | 13408 (49.6) | 7360 (55.1) |
| ATS | 50887 (59.8) | 18069 (55.5) | 6700 (57.6) | 17158 (63.4) | 8645 (64.7) |
| Other drug$ | 6579 (7.7) | 2212 (6.8) | 1057 (9.1) | 2006 (7.4) | 1248 (9.3) |
| Injects drugs | 41309 (48.5) | 14734 (45.2) | 10322 (88.7) | 9860 (36.5) | 6129 (45.8) |
| **Type of clinic** | | | | | |
| Fixed site/DIC | 74352 (87.4) | 29564 (90.8) | 9957 (85.6) | 23079 (85.3) | 11333 (84.8) |
| Mobile | 10741 (12.6) | 3003 (9.2) | 1673 (14.4) | 3961 (14.6) | 2026 (15.2) |
| **Clinic location (yes vs no)** | | | | | |

*(Continued)*

**Table 1.** (Continued)

| Variable | Total | Excluded participants* | | | Included |
|---|---|---|---|---|---|
| | | No HIV test | HIV+ | One HIV test | |
| Border area | 16627 (19.5) | 5706 (17.5) | 2083 (26.5) | 6038 (22.3) | 1751 (13.1) |
| Mining area | 64863 (76.2) | 22929 (70.4) | 9864 (84.8) | 21310 (78.8) | 10438 (78.1) |
| **Clinic State/Region** | | | | | |
| Kachin | 49925 (59.6) | 18892 (58.0) | 7907 (68.0) | 15730 (58.2) | 7183 (53.8) |
| Sagaing | 27732 (32.6) | 10456 (32.1) | 3200 (27.5) | 8741 (32.3) | 5071 (38.0) |
| Shan (North) | 6824 (8.0) | 2724 (8.4) | 480 (4.1) | 2495 (9.2) | 1105 (8.3) |

*497 (0.5%) people were excluded due to mismatch in names at registration and HIV test or if HIV test date was prior to registration date. All indicators were collected at point of client registration $ Other drugs included, marijuana, formula, diazepam, alcohol and methadone. ATS= amphetamine type stimulants DIC=drop in centre

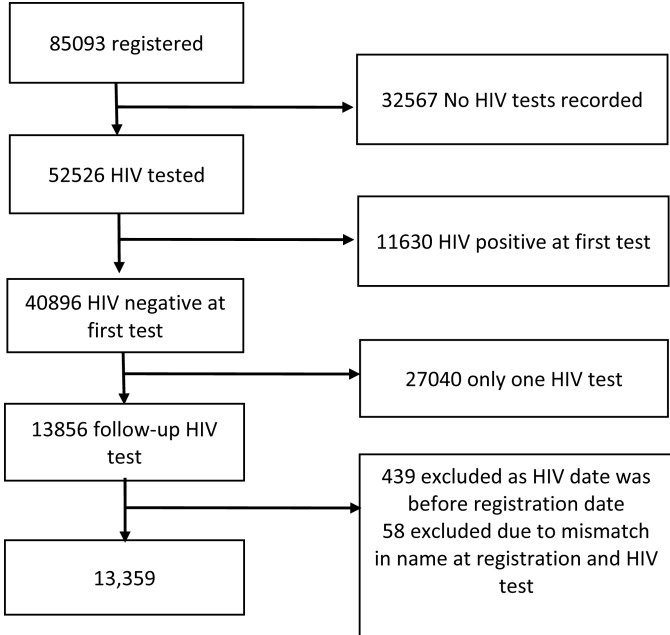

**Fig 1. HIV incidence analysis exclusions among AHRN clients in Myanmar, 2014-2021.**

fewer participants were registered in borderland areas in the final analytical sample (13.1%) compared to the total sample (19.5%). A higher proportion were excluded due to HIV positive first test (26.%) or having no follow-up HIV test (22.3%).

## Characteristics by registration in borderland or mining areas

Fig 2 depicts the geographical distribution of townships in which AHRN services operate according to location in borderland or mining areas and inclusion in the analysis. We included data from 15/22 townships, of which 8 were located in mining areas, 3 were in borderland areas, 2 were in both mining and borderland areas and 2 were in neither a borderland nor mining area. Townships were excluded where projects sites did not work with people who use drugs and one site conducted HIV testing only but did not record any data.

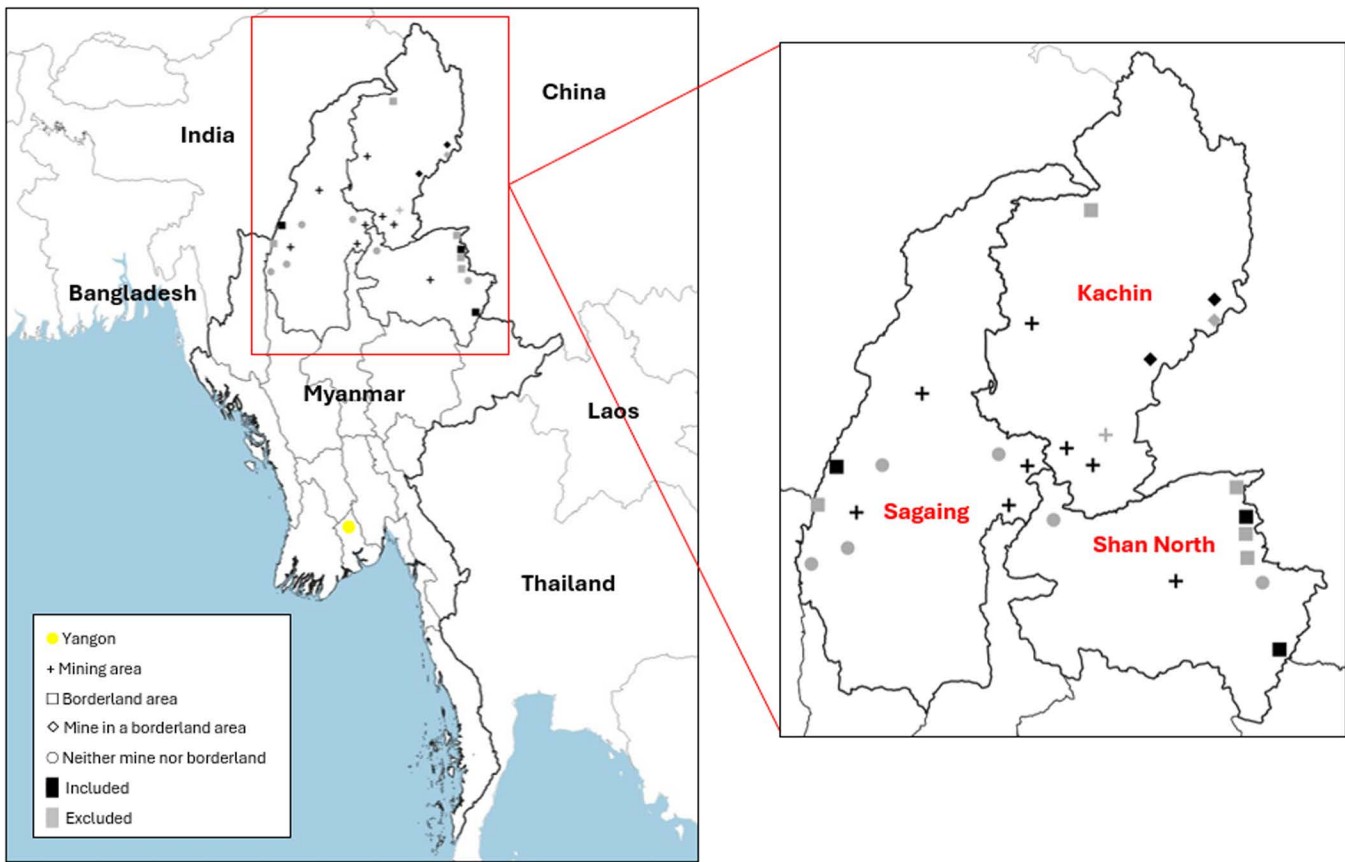

**Fig 2. Distribution of AHRN-operated-mobile or fixed site harm reduction services in borderland and mining areas.**

Republished from Myanmar Management Information Unit under a CC BY licence, with permission from the Humanitarian Data Exchange original copyright 19th June 2023.

Table 2 summarises the characteristics of clients by registration in border area vs non-border area and registration in a mining area vs non-mining area. Clients who registered at an AHRN clinic in border areas were more likely to be married (37.5%) than clients who registered elsewhere (22.9%), to be illiterate (8.3% vs 3.0%), to work in agricultural labour (52.4% vs 31.9%) to inject drugs (54.4% vs 44.4%), and were less likely to have experience of migration (6.7% vs 16.6%) or use ATS (54.8% vs 66.0%). Clients who registered at an AHRN clinic in a mining area were more likely to be female (5.6%) than clients who registered elsewhere (2.2%), less likely to use opium (50.9% vs 69.9%) and were more likely to be a migrant (18.9% vs 2.1%)

## HIV incidence rate and associations with demographic characteristics

Of the 13,359 included clients, there were 29,491 person-years of follow up (median 1.7 years, IQR 0.9–3.1) recorded between 2014 and 2021. A total of 33,022 HIV tests were conducted (median 2 IQR 2–3 tests per participant and 1.1 years between test); 1,114 clients had an HIV-positive follow-up test and another 12,245 had only HIV-negative follow up results, corresponding to an HIV incidence rate of 3.8/100 person-years. Overall, between 2014–2021 there were 12,736 men who registered at AHRN clinics, contributing 28,251 person-years of observation, 1,085 HIV seroconversions, and an HIV incidence rate of 3.8 per 100 person-years, compared to 2.3 for women.

**Table 2. Characteristics of included clients by migration experience, location in borderland or mining areas (n=13359).**

| Variable | Borderland area | | Mining area | |
|---|---|---|---|---|
| | **No** | **Yes** | **No** | **Yes** |
| Total* | 11632 | 1772 | 2919 | 10473 |
| **Age, median (IQR)** | 38 (32-46) | 36 (30-44) | 38 (31-46) | 38 (31-46) |
| **Gender** | | | | |
| Male | 11079 (95.3) | 1662 (93.8) | 2856 (97.8) | 9884 (94.4) |
| Female | 553 (4.8) | 110 (6.2) | 63 (2.2) | 589 (5.6) |
| **Education** | | | | |
| Illiterate/other | 348 (3.0) | 147 (8.3) | 77 (2.6) | 417 (4.0) |
| Primary/literate | 4088 (35.1) | 443 (25.0) | 1122 (38.4) | 3407 (32.5) |
| Up to secondary | 6166 (53.0) | 1037 (58.5) | 1519 (52.0) | 5676 (54.2) |
| Tertiary or more | 478 (4.1) | 76 (4.3) | 147 (5.0) | 406 (3.9) |
| Missing | 552 (4.8) | 69 (3.9) | 54 (1.8) | 567 (5.4) |
| **Marital** | | | | |
| Post-marital | 484 (4.2) | 73 (4.1) | 86 (2.9) | 467 (4.5) |
| Married | 2665 (22.9) | 664 (37.5) | 670 (22.9) | 2657 (25.4) |
| Single | 1495 (12.8) | 289 (16.3) | 258 (8.8) | 1520 (14.5) |
| Missing | 6988 (60.1) | 746 (42.1) | 1905 (65.3) | 5829 (55.7) |
| **Occupation** | | | | |
| Agriculture | 3716 (31.9) | 928 (52.4) | 1934 (66.3) | 2710 (25.9) |
| Driver | 484 (4.2) | 51 (2.9) | 98 (3.4) | 437 (4.2) |
| Drug/casino/sex | 233 (2.0) | 47 (2.6) | 45 (1.5) | 224 (2.1) |
| Mining | 3773 (32.4) | 26 (1.5) | 47 (1.6) | 3752 (35.8) |
| Student/no income | 561 (4.8) | 190 (10.7) | 156 (5.4) | 595 (5.7) |
| Casual | 506 (4.3) | 100 (5.6) | 194 (6.6) | 412 (3.9) |
| Skilled/ office | 238 (2.0) | 86 (4.9) | 128 (4.4) | 196 (1.9) |
| Other/ missing | 2118 (18.2) | 343 (19.4) | 315 (10.8) | 2145 (20.5) |
| **Experience of migration** | | | | |
| No | 4573 (39.3) | 773 (43.6) | 1446 (49.5) | 3900 (37.2) |
| Yes | 1935 (16.6) | 119 (6.7) | 61 (2.1) | 1981 (18.9) |
| Missing | 5124 (44.0) | 880 (49.7) | 1412 (48.4) | 4592 (43.9) |
| **Drug use** | | | | |
| Heroin (yes vs no) | 11241 (96.6) | 1650 (93.1) | 2850 (97.6) | 10041 (95.9) |
| ATS (yes vs no) | 7672 (66.0) | 972 (54.8) | 2108 (72.2) | 6524 (62.3) |
| Opium (yes vs no) | 6502 (55.9) | 873 (49.3) | 2038 (69.9) | 5337 (50.9) |
| Other (yes vs no) | 1248 (10.7) | 82 (4.6) | 116 (4.0) | 1214 (11.6) |
| Injects drugs(yes vs no) | 5168 (44.4) | 964 (54.4) | 1266 (43.4) | 4866 (46.5) |
| **Clinic location** | | | | |
| **State/Region** | | | | |
| Kachin | 6214 (53.4) | 999 (56.4) | 0 (0.0) | 7213 (68.9) |
| Sagaing | 4541 (39.0) | 540 (30.5) | 2699 (92.4) | 2383 (22.7) |
| Shan (North) | 876 (7.5) | 233 (13.1) | 221 (7.6) | 876 (8.4) |
| **In border area** | | | | |
| No | | | 2157 (73.9) | 9475 (90.5) |
| Yes | | | 762 (26.1) | 998 (9.5) |
| **In mining area** | | | | |
| No | 2157 (18.5) | 762 (43.0) | | |
| Yes | 9475 (81.5) | 998 (56.3) | | |

*(Continued)*

**Table 2.** (Continued)

| Variable | Borderland area | | Mining area | |
|---|---|---|---|---|
| | No | Yes | No | Yes |
| Missing | 0 (0.0) | 12 (0.7) | | |

*Missing data: Migrants (n=6007, 44.8%), Borderland (n=1771,13.2%), Mining (n=12 0.1%).

The incidence rate ratio (IRR) for women compared to men was 0.61 (95% CI of 0.38–0.97). We also observed lower HIV incidence among clients who were older compared to those <=25 years (IRR>=45 years 0.17 95% CI 0.12–0.25; 35–44 years= 0.40 95% CI 0.28–0.57; 25–34 years=0.65 95% CI 0.48–0.87), migrants compared to non-migrants (IRR 0.55 95% CI 0.37–0.82) or married compared to widowed/divorced (IRR 0.63 95% CI 0.46–0.86). We observed greater HIV incidence among clients with higher educational attainment, who used heroin compared to those who did not (3.1 95% CI 2.27–4.27), used drugs via injection versus non-injecting (IRR 5.5 95% CI 4.25–7.04), or were registered at a clinic in a border area versus not (IRR 1.6 95% CI 0.97–2.62). Results are summarised in Table 3.

### Association between HIV incidence and location in a borderland or mining area

After adjusting for potential confounders, clients who were registered at a clinic in a border area had 67% higher incidence of HIV (IRR 1.67, 95% CI 1.13–2.45) relative to those who registered elsewhere. We did not observe a difference in the incidence of HIV for clients according to whether their clinic registration was in a mining area or not. (see Table 4).

### Discussion

Our study found high HIV incidence among people who use drugs accessing services through AHRN of 3.8 cases per 100 person-years and higher among those who inject (6.8/100 pyrs) and those aged 25 years or younger (8.9/100 pyrs). Higher seroconversion rates were observed among those registered in a border area compared to those who were not (IRR 1.67 95% CI 1.13–2.45) and there was no evidence of association with location in a mining area.

Findings support evidence of differential risk of HIV transmission among people who use drugs situated in border areas than adults in non-border areas [24,30–32]. Research among people who inject heroin in borderland areas in remote parts of Shan State document frequent injection (at least daily) and high levels of sharing of needles/syringes [5]. Other evidence from Ruili immediately across the border in China, found that injecting drugs in both China and Myanmar was associated with increased odds of sharing injecting equipment and weaker evidence of an association with testing positive for HIV [19]. Elevated ATS use in borderland cities in proximity to ATS distribution has been observed both in the region and in border cities between Mexico and the United States [30,33]. We found some differences in drug use between those located in borderland areas compared to non-borderland, ATS use was less frequently reported but a greater proportion of people injected drugs, had higher levels of illiteracy and worked in agriculture. This likely reflects the increased availability of heroin as well as poorer access to education and more limited employment opportunities in borderland areas. The physical landscape (thick forests and mountains), historic armed conflict, insecure income and involuntary migration make it difficult accessing health services and other necessary infrastructures [34]. Proportionally fewer AHRN clients were registered at project sites in borderland areas (19.5%) compared to mining areas (76.2%). Assuming demand for services

**Table 3. Client characteristics and crude association with HIV incidence rate among AHRN clients who use drugs, 2014-2022 (n=13,359).**

| Variable | incident HIV | Total at start of observation | Person-years of observation | Rate/100 pyrs* | Incidence Rate Ratio (95%CI) [a] |
|---|---|---|---|---|---|
| **Total** | 1114 | 13359 | 29491 | 3.8 | |
| **Age** | | | | | |
| <=25 | 87 | 693 | 981 | 8.9 | 1.0 |
| 25-34 | 502 | 4307 | 8698 | 5.3 | 0.65 (0.48, 0.87) |
| 35-44 | 390 | 4655 | 10964 | 3.2 | 0.40 (0.28, 0.57) |
| >=45 | 135 | 3740 | 8847 | 1.3 | 0.17 (0.12, 0.25) |
| **Gender** | | | | | |
| Male | 1085 | 12736 | 28251 | 3.8 | 1.0 |
| Female | 29 | 623 | 1240 | 2.3 | 0.61 (0.38, 0.97) |
| **Education** | | | | | |
| Illiterate/other | 30 | 498 | 1140 | 2.6 | 1.0 |
| Primary/literate | 350 | 4511 | 9432 | 3.7 | 1.39 (1.06, 1.89) |
| Up to secondary | 679 | 7173 | 16651 | 4.0 | 1.50 (1.19, 1.96) |
| Tertiary or more | 44 | 556 | 1370 | 3.2 | 1.21 (0.83, 1.80) |
| Missing | 28 | 621 | 898 | 2.2 | 0.83 (0.40, 1.76) |
| **Occupation** | | | | | |
| Agriculture | 376 | 4631 | 8665 | 4.3 | 1.0 |
| Driver | 52 | 536 | 1438 | 3.6 | 0.83 (0.65, 1.06) |
| Drug/casino/sex | 18 | 270 | 611 | 2.9 | 0.68 (0.37, 1.22) |
| Mining | 310 | 3798 | 9253 | 3.3 | 0.77 (0.51, 1.17) |
| Students/ no income | 77 | 735 | 1779 | 4.3 | 1.00 (0.77, 1.29) |
| Casual | 53 | 602 | 1375 | 3.8 | 0.89 (0.67, 1.18) |
| Skilled/ office | 22 | 326 | 719 | 3.0 | 0.9 (0.46, 1.05) |
| Other/ missing | 206 | 2461 | 5651 | 3.6 | 0.84 (0.66, 1.06) |
| **Marital status** | | | | | |
| Post-marital | 53 | 555 | 993 | 5.3 | 1.0 |
| Married | 191 | 3293 | 5705 | 3.3 | 0.63 (0.46, 0.86) |
| Single | 147 | 1780 | 3027 | 4.8 | 0.91 (0.73, 1.13) |
| Missing | 723 | 7731 | 19765 | 3.6 | 0.68 (0.42, 1.11) |
| **Experience of migration** | | | | | |
| No | 388 | 5319 | 9196 | 4.2 | 1.0 |
| Yes | 94 | 2043 | 4014 | 2.3 | 0.55 (0.37, 0.82) |
| Missing | 632 | 5997 | 16280 | 3.9 | 0.92 (0.64, 1.33) |
| **Uses heroin** | | | | | |
| No | 11 | 462 | 887 | 1.1 | |
| Yes | 1103 | 12897 | 28604 | 3.8 | 3.11 (2.27, 4.27) |
| **Uses opium** | | | | | |
| No | 461 | 5999 | 12673 | 3.6 | 1.0 |
| Yes | 653 | 7360 | 16818 | 3.9 | 1.07 (0.89, 1.28) |
| **Uses ATS** | | | | | |
| No | 404 | 4714 | 10963 | 3.7 | 1.0 |
| Yes | 710 | 8645 | 18529 | 3.8 | 1.04 (0.86, 1.25) |
| **Uses injection drugs** | | | | | |
| No | 198 | 7230 | 15976 | 1.2 | 1.0 |
| Yes | 916 | 6129 | 13516 | 6.8 | 5.5 (4.25, 7.04) |
| **Location of clinic** | | | | | |
| Kachin | 618 | 7183 | 17598 | 3.5 | 1.0 |

*(Continued)*

**Table 3.** (Continued)

| Variable | incident HIV | Total at start of observation | Person-years of observation | Rate/100 pyrs* | Incidence Rate Ratio (95%CI) [a] |
|---|---|---|---|---|---|
| Sagaing | 421 | 5071 | 9348 | 4.5 | 1.28 (0.87, 1.88) |
| Shan (North) | 75 | 1105 | 1030 | 2.9 | 0.84 (0.56, 1.25) |
| **In border area** | | | | | |
| No | 916 | 11608 | 25978 | 3.5 | 1.0 |
| Yes | 198 | 1751 | 3514 | 5.6 | 1.60 (0.97, 2.62) |
| **In mining area** | | | | | |
| No | 259 | 2909 | 6427 | 4.0 | 1.0 |
| Yes | 855 | 10438 | 23054 | 3.7 | 0.92 (0.68, 1.24) |
| **Registration year** | | | | | |
| 2014 | 125 | 779 | 3640 | 3.4 | 1.0 |
| 2015 | 123 | 929 | 3947 | 3.1 | 0.90 (0.67, 1.23) |
| 2016 | 147 | 1571 | 5593 | 2.6 | 0.76 (0.51, 1.14) |
| 2017 | 159 | 1565 | 4213 | 3.8 | 1.10 (0.74, 1.64) |
| 2018 | 227 | 2268 | 4571 | 5.0 | 1.44 (0.94, 2.21) |
| 2019 | 212 | 3312 | 4861 | 4.3 | 1.27 (0.75, 2.13) |
| 2020 | 116 | 2642 | 2492 | 4.7 | 1.35 (0.83, 2.20) |
| 2021 | 5 | 293 | 173 | 2.9 | 0.84 (0.29, 2.45) |

[a]Crude incidence rate ratio estimated with Poisson regression with log survival time as an offset and 95% CI adjusted for clustering by project site.

**Table 4.** Geographic characteristics and adjusted association with HIV incidence among AHRN clients, 2014-2022.

| Risk factor | Seropositive cases/Total | Adjusted Incidence Rate Ratio (95% CI) [a] | P value |
|---|---|---|---|
| **Client in border area clinic** | 198/11608 | 1.67 (1.13, 2.45)[b] | 0.009 |
| **Client in mining area clinic** | 855/10438 | 0.93 (0.72, 1.20)[c] | 0.582 |

[a]Incidence rate ratio estimated with Poisson regression with log survival time as an offset

[b]IRR adjusted for adjusted for age, gender, education, calendar year of registration, registration in mining area, state of clinic

[c]IRR adjusted for adjusted for age, gender, education, calendar year of registration

All 95% CIs adjusted for clustering by project site.

is comparable, this could indicate reduced access to harm reduction services that, alongside increased levels of injecting, and may contribute to elevated incidence observed in borderland areas but not mining areas. Our findings suggest less ATS use and comparable prevalence of heroin or injecting drugs in mining areas compared to non-mining areas in contrast to reports that document intensive ATS and heroin use among people working in mines [18]. Given difficulties in providing services in sensitive areas such as borderlands and mines, further research is needed to better quantify existing coverage of harm reduction services in these areas, characterise the population in need and inform the immediate scale up of HIV prevention and treatment services [18].

Our estimates of incidence are in line with analyses of programmatic data from Médecins du Monde (MDM) providing harm reduction services in Kachin State that estimated incidence to be 7.1/100 pyrs (n=2277) [9]. Our incidence of 8.9/ 100 pyrs among people aged 25 years or younger is comparable with estimates among a sample of similarly aged men who have sex with men and transgender populations recruited in Myanmar and Mandalay (77% aged 25 years or younger) [35]. We found no evidence that incidence among clients declined over time in contrast to evidence from prospective cohorts of people who inject

drugs in Thailand and the MDM programmatic data [9,36]. Our analyses did not account for the provision of OAT or needle/syringes although both interventions are a cornerstone of AHRN's HIV prevention activities and both OAT and needle/syringe provision are associated with reduced HIV incidence in Kachin and globally [7,9]. The higher incidence associated with injecting (6.8/100 pyrs) suggest the HIV epidemic is driven primarily through injecting risk practices, but incidence was still high (1.2/100 pyrs) among people using drugs via non-injecting routes. This points to the need to prioritise sexual risk reduction interventions at harm reduction programmes.

The strength of our analysis lies in the large longitudinal sample of harm reduction clients across a wide geographic area. Our findings provide the first estimate of HIV incidence in Myanmar among people who use drugs that focusses on the examination of social, political and physical context in the form of borderland and mining areas, building on the substantial evidence of the utility of programme data to estimate HIV incidence and illustrating their use in measuring structural factors in transmission [29,37]. A key limitation is the measurement of exposure variables only at first registration, that does not necessarily reflect exposure at seroconversion. In relation to borderland and mining areas, individuals may have moved on by the time they engage in a second HIV test. Our measure of mining reflects risk in the wider community proximal to mines, rather than occupational hazards of working in mines or related activities. Despite increased drug use among people working in mines, HIV incidence among people reporting mining as an occupation is comparable with those registered in mining areas [5]. This finding supports other evidence identifying industrial mines as hotspots for HIV transmission triggering changes in practices of local communities more conducive to HIV transmission due to migration of people for short term work from areas of high HIV prevalence, and greater propensity for people to engage with sex work or other risky sexual practices [38]. While our findings don't suggest that the presence of mining activity substantially alters HIV vulnerability in the wider community of people using drugs, we do observe a greater proportion of women and migrants in mining areas suggestive of changes to the community. Appropriate HIV responses need to be tailored to address the needs of these populations.

Analyses draw on a convenience sample from AHRN project sites and may not represent the overall population of people using drugs, particularly given over a third were excluded due to not having an HIV test. Our sample is similar to other evidence from Myanmar including a multi-site community recruited sample of people who inject drugs conducted in 2017 who were also predominantly male (95.6% vs 98.2%) and used heroin (93.9% vs 98.8%) [10]. Although our sample was older (median 36 vs 30 years) than this and other surveys [10,39,40] and with lower prevalence of illiteracy (4.6% vs 15.7%) [40]. The National AIDS Programme guidelines recommend HIV testing every 6 months, but the median time between tests was 1.1 years among our sample with 38.3% of clients not being tested at all and only 26.4% with repeat tests. While this is low, it is in line with community surveys that suggest 51.6% of people who inject drugs had never had an HIV test and 48% were tested over one year ago [10]. Nevertheless our findings point to the need to increase uptake of repeat testing among this population and particularly among young people (age <25 years) who were less represented in the analytical sample due to having fewer repeat tests and testing positive at first test. This might have also resulted in an underestimate of HIV incidence. Gender was measured as a binary, so our analyses fail to document additional risk among transgender populations. Only 4% of the sample were female. Further work is needed to engage women who use drugs in these services and document prevention and treatment needs [5,10]. Our reliance on routine programmatic data with a limited number of indicators limits our understanding of the role of other structural factors or mediators, such as ethnicity for example that might contribute to conflict in borderland

areas and HIV transmission. Missing data and lack of linkage to measures of OAT and ART uptake means it was not possible to control for all potential confounders in the analysis.

## Conclusions

Our findings contribute to the body of evidence that document the importance of borderland areas in disease transmission and the imperative to intensify harm reduction interventions with a focus on cross-border interventions. Evaluation of cross-border interventions involving needle/syringe distribution and peer education points to their effect in reducing HIV incidence among people who inject drugs in China and Vietnam [31]. HighHIV incidence and insufficient coverage of these key interventions clearly point to the need to expand uptake and coverage of NSPs and OAT alongside use of pre-exposure prophylaxis and access to HIV treatment among people who inject drugs, ensuring interventions are delivered in borderland areas. The recent introduction of PrEP in Myanmar is welcome, but current pilot studies in harm reduction services are focussed on people who have injected in the last 6 months only, excluding people on OAT or using drugs via other routes. Our findings highlight the importance of extending eligibility criteria to include people who use drugs more broadly, including those with multiple partners or engaging in unprotected sex in areas of high HIV incidence or prevalence in line with national guidelines for other key populations. Increasing uptake of HIV testing is imperative alongside the scale up of evidenced based interventions to address sexual and injecting risk practices. These interventions are essential to curb the high rates of HIV transmission among these populations particularly among young people and those in remote borderland areas.

## Acknowledgement

We thank AHRN clients and staff for participation in the study.

## Author contributions

**Conceptualization:** Lucy Platt, Khine Wut Yee Kyaw, Sujit D. Rathod, Bayard Roberts.

**Data curation:** Khine Wut Yee Kyaw, Sujit D. Rathod, Aung Yu Naing.

**Formal analysis:** Lucy Platt, Khine Wut Yee Kyaw, Sujit D. Rathod, Sophia Garkov.

**Methodology:** Bayard Roberts.

**Software:** Sophia Garkov.

**Writing – original draft:** Lucy Platt, Sujit D. Rathod.

**Writing – review & editing:** Lucy Platt, Khine Wut Yee Kyaw, Sujit D. Rathod, Aung Yu Naing, Sophia Garkov, Murdo Bijl, Bayard Roberts.

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
