## [Decision Letter · Decision Letter 0]

4 Mar 2024

PONE-D-23-37790The effect of migration, location in mining or borderland areas on HIV incidence among people who use drugs attending a harm reduction programme in Myanmar, 2014-2021: a retrospective cohort studyPLOS ONE

Dear Dr. Platt,

Thank you for submitting your manuscript to PLOS ONE. After careful consideration, we feel that it has merit but does not fully meet PLOS ONE’s publication criteria as it currently stands. Therefore, we invite you to submit a revised version of the manuscript that addresses the points raised during the review process.

We regret the time it took to get this paper reviewed.  As academic editor, I undertook the second review, as it is in one of my areas of expertise. 

lease submit your revised manuscript by Apr 18 2024 11:59PM.  If you will need more time than this to complete your revisions, please reply to this message or contact the journal office at plosone@plos.org . Please include the following items when submitting your revised manuscript:

We look forward to receiving your revised manuscript.

Kind regards,

Kimberly Page, PhD, MPH

Academic Editor

PLOS ONE

Journal Requirements:

"This work is supported by the ESRC, Drugs and (dis)order: Building sustainable peacetime economies in the aftermath of war, UKRI Award no. ES/P011543/1, 2017–2021, as part of the Global Challenges Research Fund. The funder had no role in the design of the study, collection, analysis, interpretation of data and writing of the manuscript."

5. We note that Figure 2 in your submission contain map image which may be copyrighted. All PLOS content is published under the Creative Commons Attribution License (CC BY 4.0), which means that the manuscript, images, and Supporting Information files will be freely available online, and any third party is permitted to access, download, copy, distribute, and use these materials in any way, even commercially, with proper attribution. For these reasons, we cannot publish previously copyrighted maps or satellite images created using proprietary data, such as Google software (Google Maps, Street View, and Earth). For more information, see our copyright guidelines: http://journals.plos.org/plosone/s/licenses-and-copyright.

Additional Editor Comments:

This paper presents analyses of programmatic data collected by Harm Reduction Service agencies in Myanmar, and estimates HIV incidence from repeat testers. I appreciate two sided coin that the data bring: rich, locally collected, large numbers, but also the weaknesses, mostly in the realm of missing data. I concur with Reviewer #1's notes and critiques. I add that I believe that the paper could use more focus. The Introduction starts with the world (all over the place!), and takes us to some small villages in Myanmar. The authors have not really given us a description of the similarities and differences between mining and borderland areas. What is the rationale behind this categorization and exploration of HIV incidence? The reader would benefit from this information. What is not known in this context re: HIV risk and incidence.? I appreciate that migration is "interesting" but I fear the weaknesses of that data may outweigh their usefulness. The authors seem to be trying to do too much with very limited data on the key exposure. I think a reformulation focused on the robust measures they have would be of interest. In the end, these are a LOT of comparisons and the picture at the end is muddy. A sharper focus throughout the paper will be more informative. I also urge the authors to use consistent language for example: Amphetamine, or ATS is used interchangeably. Programmatic data have been used for a long time for HIV incidence estimation, so this is not novel. The authors do say that this is novel for "structural" data. But the structural - in this instance is really occupational or geographic.

Reviewers' comments:

Reviewer's Responses to Questions

**Comments to the Author**

1. Is the manuscript technically sound, and do the data support the conclusions?

Reviewer #1: Partly

2. Has the statistical analysis been performed appropriately and rigorously?

Reviewer #1: Yes

3. Have the authors made all data underlying the findings in their manuscript fully available?

Reviewer #1: No

4. Is the manuscript presented in an intelligible fashion and written in standard English?

Reviewer #1: Yes

5. Review Comments to the Author

Reviewer #1: This is a very interesting paper that seeks to build on recent work on HIV incidence in Myanmar using routinely collected service data and focus on the distribution of cases across geographic exposures. The incidence figures largely replicate those reported using service data from another group of service provider. The main finding in relation to the geographic markers included is that the group of people identifying as having moved away from home at first service contact had a lower incidence of HIV than those who did not.

The analysis strategy is somewhat difficult to follow but my main concern is with the geographic markers included. Having a contact with a service in a border area seems relatively clear, assuming that the service catchment and the area covered by outreach services are relatively local. In relation to ‘mining’, the marker is far less clear. It appears that service contact in a mining area is taken as a proxy for being involved in mining but it really refers to the macro-characteristics of the area rather than mining per se. This seems to be pretty weak measurement. I’m not sure why the occupation data were not used as a more direct measure of ‘mining’. Although the issue of missing data is apparent, this also applies to the migration measure. On this point, it seems as though the migration measure includes a large amount of missing data – how was this managed in the regression analysis? It looks to me like it was simply ‘yes’ versus ‘non’ with missing data excluded. Given the large amount of missing data I’m not sure this approach is appropriate, especially given the argument made for not taking the same approach in relation to mining occupation data. Overall, I think a simpler focus on border areas service provision alone is probably warranted with the other geographic measures mentioned as avenues for further investigation.

The labelling of variables needs to be made consistent throughout –e.g., ‘uses injection drugs’ appears equated with ‘injection drug use’. Similarly, ‘left home’ is equated with migrated which could simply mean leaving a parent’s residence rather than how migration is typically understood. These are different measures and if equated this should be specified. Finally, I’m not sure that the PWID/PWUD acronyms are particularly useful or person--centred. I would suggest spelling these out in full.

The figure including the map of the areas involved was difficult to follow (labelled twice too). Given Yangon is not a mining or border area, why not limit the figure to a higher resolution of the relevant areas?

The discussion covers issues not really canvassed in the study (e.g. speculation about demand in different areas with no real supporting evidence [p.18]). A sharper focus on the relation to the MDM work mentioned in the discussion and how this study’s findings build on this would suffice.

6. PLOS authors have the option to publish the peer review history of their article (what does this mean? ). If published, this will include your full peer review and any attached files.

**Do you want your identity to be public for this peer review?** For information about this choice, including consent withdrawal, please see our Privacy Policy .

Reviewer #1: No

---

## [Author Response · Author response to Decision Letter 1]

28 Jun 2024

We thank the reviewer for their comments. As described in our response above, we have tried to better justify our focus on location in borderland and mining areas. In line with other evidence we are taking defined geographical areas with specific political, physical and social attributes to represent important parts of the risk environment shaping HIV risk behaviours. Borderland areas are situated along the borders with China and India with the presence of cross-border trade and movement, and specific socio-political characteristics, including the presence of ethnic armed groups and military activity. Mining areas are identified based on the presence of significant mining activities, particularly jade, gold mining, the concentration of mining operations, and the socio-economic impact of mining on the local population. As we describe in the methods, location in borderland or mining areas was extracted from profiles of townships, crossed checked with project staff. AHRN staff conduct comprehensive mapping exercises to delineate the specific boundaries of borderland and mining areas. Regular updates to classifications are made based on changing conditions and new information from ongoing. We note that 2/15 townships were located in both mining and borderland areas. We have added in this detail to better describe the approach in the methods.

The analysis was set up to examine the macro-contexts of the mining area and not as a proxy for occupation in a mine. We hope that our additions to the introduction have better justified this approach. All of the people reporting mining as their occupation were located in mining areas. We also note no difference in HIV incidence between those reporting mining as an occupation (3.3 per 100 pyrs) and those located in mining areas (3.7 per 100 pyrs). We have added a sentence into the discussion of limitations reflecting on this point.

We agree that the issue of missing data in our measurement of migration is problematic and for that reason as well as to better focus the analysis we have removed the migration measure from the analysis. We have tightened up our labelling to ensure it is consistent and revised the use of acronyms related to drug use in order to ensure they are person-centred. We have clarified that our definition of migration is living away from hometown and movement to various locations for three months or more.

We feel it’s important to include a map of the whole country for two reasons. Firstly, we feel that for an international readership it is important to demonstrate the scale of the country and the distance from the study areas to the capital. Secondly the aim is to show the geographic distribution of the sites by borderland and mining categories without revealing the location of sites in order to reduce to maximise confidentiality of participants.

We have revised the discussion to better link with the results presented and to build on the findings of the analysis of MDM data to estimate incidence. Given that this is the first estimate of incidence among people using drugs via non-injecting drugs it is appropriate however to comment on the implications for interventions for that this population.

---

## [Editor Report · Decision Letter 1]

24 Jul 2024

PONE-D-23-37790R1The effect of location in mining or borderland areas on HIV incidence among people who use drugs attending a harm reduction programme in Myanmar, 2014-2021: a retrospective cohort studyPLOS ONE

Dear Dr. Platt,

Thank you for submitting your manuscript to PLOS ONE. After careful consideration, we feel that the revisions are very responsive but the manuscript would benefit from some minor edits to meet PLOS ONE’s publication criteria.  Therefore, we invite you to submit a revised version of the manuscript that addresses the points raised during the review process.

We look forward to receiving your revised manuscript.

Kind regards,

Kimberly Page, PhD, MPH

Academic Editor

PLOS ONE

Journal Requirements:

Additional Editor Comments:

The authors have been very responsive to comments and the paper tells a more focused story. Some minor edits will contribute to further clarity and interpretation.

1. Table 1 heading currently says: Table 1. Characteristics of newly registered AHRN clients by HIV incidence analysis inclusion status, Myanmar, 2014-2022. I suggest this be retitled to take out "incidence analysis' - its just confusing. Its really those who met inclusion or exclusion criteria. This table is quite dense and would also benefit from some alignment.

2. I understand the justification for the large map as provided in responses. However, it is not compelling and the sites are really hard to distinguish. We still feel that a map with more resolution would be more helpful. Have you considered the large one but adding an "insert" or window view of the specific area?

3. Table 3: change person-years of survival to person-years of observation.

4. Discussion: Please add to the sentence "Higher seroconversion rates were observed among those registered in border area ..... relative to <what group="">.

5. Consider discussing the impact of missing tests in the younger group that had high incidence and the potential for underestimation in your limitations paragraph.

While revising your submission, please upload your figure files to the Preflight Analysis and Conversion Engine (PACE) digital diagnostic tool, https://pacev2.apexcovantage.com/ . PACE helps ensure that figures meet PLOS requirements. To use PACE, you must first register as a user. Registration is free. Then, login and navigate to the UPLOAD tab, where you will find detailed instructions on how to use the tool. If you encounter any issues or have any questions when using PACE, please email PLOS at figures@plos.org . Please note that Supporting Information files do not need this step.</what>

---

## [Author Response · Author response to Decision Letter 2]

30 Aug 2024

We have edited the title as requested. We have attempted to align the table to make it more readable and removed the date of registration to make it less dense.

We agree that a map with more resolution would be more useful. We have added an insert magnifying the study area in the context of the region, so that the distribution of sites is clearer.

As a consequence of the additional work to revise the map we have added Sophia Garkov as an additional author.

We have made the minor edits requested to Table 3 and the Discussion.

We thank the reviewer for this suggestion and agree this is an important point to highlight. We have added in a sentence into the limitations raising the need to prioritise testing among younger age groups. We also note that the lower uptake of repeat testing and higher prevalence of HIV may result in less representation of this age group in our analytical sample leading to an underestimation of incidence.

---

## [Editor Report · Decision Letter 2]

3 Sep 2024

The effect of location in mining or borderland areas on HIV incidence among people who use drugs attending a harm reduction programme in Myanmar, 2014-2021: a retrospective cohort study

PONE-D-23-37790R2

Dear Dr. Platt,

We’re pleased to inform you that your manuscript has been judged scientifically suitable for publication and will be formally accepted for publication once it meets all outstanding technical requirements. Thank you for your attention to these last comments. Congratulations!

Kind regards,

Kimberly Page, PhD, MPH

Academic Editor

PLOS ONE
---

## [Editor Report · Acceptance letter]

PONE-D-23-37790R2

PLOS ONE

Dear Dr. Platt,

I'm pleased to inform you that your manuscript has been deemed suitable for publication in PLOS ONE. Congratulations! Your manuscript is now being handed over to our production team.

Kind regards,

on behalf of

Dr. Kimberly Page

Academic Editor

PLOS ONE